# Reverse-engineering NLI: A study of the meta-inferential properties of Natural Language Inference

**Rasmus Blanck & Bill Noble**

The Centre for Linguistic Theory and Studies in Probability (CLASP),
Department of Philosophy, Linguistics and Theory of Science,
University of Gothenburg
{rasmus.blanck,bill.noble}@gu.se

**Stergios Chatzikyriakidis**
Computational Linguistics and Language Technology Lab (UCRC),
Department of Philology,
University of Crete
stergios.chatzikyriakidis@uoc.gr

## Abstract

Natural Language Inference (NLI) has been an important task for evaluating language models for Natural Language Understanding, but the logical properties of the task are poorly understood and often mischaracterized. Understanding the notion of inference captured by NLI is key to interpreting model performance on the task. In this paper we formulate three possible readings of the NLI label set and perform a comprehensive analysis of the meta-inferential properties they entail. Focusing on the SNLI dataset, we exploit (1) NLI items with shared premises and (2) items generated by LLMs to evaluate models trained on SNLI for meta-inferential consistency and derive insights into which reading of the logical relations is encoded by the dataset.

## 1 Introduction

Natural Language Inference (NLI) has provided important benchmarks for assessing the Natural Language Understanding (NLU) capacity of computational models. The importance of these benchmarks comes from the theoretical role of inference in linguistic semantics, wherein what can be inferred from a given sentence is determined by its meaning. The function learned by an NLI model takes two natural language sentences as input and returns a label from a set of possible inferential relations between those two sentences. The fact that the two inputs are of the same type creates opportunities to investigate the "internal consistency" of the model; that is, the properties of the learned function in relation to how we think it should theoretically behave.

In a number of papers, the internal consistency of NLI models is investigated by means of their meta-inferential properties. For example, if we believe that contradiction is symmetric ("$a$ contradicts $b$" is equivalent with "$b$ contradicts $a$"), then we may demand that any NLI model labeling the sentence pair $(a, b)$ as a contradiction must label the pair $(b, a)$ in the same way, in order to be internally consistent with regards to our external understanding of contradiction. Here, the relation $Cab$ that holds when $a$ contradicts $b$ is an example of an *inference* relation, wheras the postulate that $Cab \rightarrow Cba$ is a *meta-inferential* relation.

A problem with many logical approaches to NLI in the literature is that (1) the NLI relations are left undefined so that it is not clear what the relations are or in which logic they can be expressed (if at all); and (2) it is not made clear from which logic the meta-inferential intuitions are coming. This is problematic since which meta-inferential relations that hold between inference relations depends on the definition of the inference relations.

In this paper, we first review related work, partly in order to start disentangling different interpretations of the NLI relations (Sec. 3), then establish a framework for analyzing the NLI relations (Sec. 4), and moreover propose three different readings of those relations, analyze their meta-inferential properties, and hypothesize that one of the readings better captures the mode of inference that is implicit in the NLI dataset. We generate new data (Sec. 5) and experimentally confirm its suitability for testing the main hypothesis (Sec. 6).[1]

## 2  Background

The SLNI corpus (Bowman et al., 2015), and its derivatives (such as MNLI (Williams et al., 2018), e-SNLI (Camburu et al., 2018), MedNLI (Romanov & Shivade, 2018), XNLI (Conneau et al., 2018), ANLI (Nie et al., 2020)), has been used extensively as a proxy for the NLI task. This is a three-way classification task, where sentence pairs $(a, b)$ of a premise $a$ and a hypothesis $b$ are to be classified using one of the labels Entailment, Neutral and Contradiction.[2] SNLI premises are all captions from the Flickr30k corpus (Young et al., 2014), and the hypotheses are constructed by human workers given the following instructions:

> We will show you the caption for a photo. We will not show you the photo.
> Using only the caption and what you know about the world:
> - Write one alternate caption that is **definitely** a **true** description of the photo. [. . . ]
> - Write one alternate caption that **might be** a **true** description of the photo. [. . . ]
> - Write one alternate caption that is **definitely** a **false** description of the photo. [. . . ]
>
> (Bowman et al., 2015, p. 634, emphasis in original)

Williams et al. (2018) give similar instructions for constructing the MNLI corpus, along with the gloss that the hypothesis of an entailment is "necessarily true or appropriate whenever the premise is true", a contradiction "necessarily false or inappropriate whenever the premise is true" and neutral "neither condition applies". This suggests that the three-way classification should cover the whole space of sentence pairs, and that neutral pairs are neither entailments nor contradictions – sentiments that are repeated by Nie et al. (2020).

Models trained on SNLI-style datasets learn to classify sentence pairs according to the NLI labels. The classification induces relations $E$, $C$, $N$ that hold on sentence pairs classified by the respective label. Externally, our logical intuition is the norm according to which we expect the models to behave. But, a salient question is whether we can actually expect the models to behave according to that norm, given the data from which the inference relations are induced. In other words, what inference relations could possibly be induced from the data provided to the models? If those relations are not compatible with our intuitions, then perhaps the models are not so much to blame as the datasets.

A preliminary inspection shows that the NLI relations are not definable in propositional logic (in a strong sense, see Sec. 3.2). The classical material conditional is also well known to be inadequate for formalizing natural language conditionals (see, e.g., Lewis & Langford 1959). Moreover, the NLI annotator instructions exhibit strong modal language ("necessary"/"definitely true"), which suggests that the NLI relations could lend themselves to a modal treatment. The strict conditional of Lewis & Langford (1959) is definable in a modal language, and is better suited for formalizing natural language conditionals, even though it suffers from shortcomings of its own when it comes to counterfactual conditionals.

There is another crucial observation to be made: Every SNLI premise comes along with the implicit assumption that the premise is *possible* – again a glaringly modal notion. The reason is that each SNLI premise is a caption of a photo of "everyday activities, events and

---

[1]For data and experimental code see: `https://github.com/GU-CLASP/reverse-engineering-NLI`

[2]This three-way classification appears as early as with the FraCaS test suite (Cooper et al., 1996), and became standard with RTE-4 (Giampiccolo et al., 2009).

scenes". If the caption is actually accurate, then it is describing a situation that is indeed possible. We may understand the existence of the photo as implying that there is a model satisfying the caption, namely the situation that the photo is a photo of. This means that the data, if constructed precisely according to the instructions, do not contain any examples of vacuously true entailments that, for example, classical propositional logic would validate. A model trained on such data is, for example, unlikely to learn the principle of explosion, and would therefore not be compatible with certain classical modes of inference.

Our approach is to define the NLI relations in a modal language that can adequately express the notions used in the NLI instructions. We consider two such readings, as well as a reading based on the "default" material conditional. The various meta-inferential principles appearing in the literature are scrutinized through this lens. We observe that certain "obvious" meta-inferential principles are valid on some of the readings but not on others. We leverage this observation to show to what degree different readings of the NLI relations are compatible with the predictions of models trained on NLI data.

## 3 Related work

### 3.1 Modalities and non-contradictory premises

Gubelmann et al. (2024) take note of the modal language ("definitely"/"necessary") of Bowman et al. (2015) and Williams et al. (2018), but without explicitly recognizing it as being modal. Instead, they conclude that the SNLI and MNLI are best "conceived as aiming at deductive validity". They leave unspecified what notion of deductive validity to consider.

Holliday et al. (2024) investigate the reasoning capabilities of LLMs on linguistic material containing conditionals and modal operators. Their approach is different from ours, as they are concerned with modalities in the object language and the relations between sentences containing such linguistic material, whereas we are using the language of modal logic as a metalanguage in which to reason about inference relations.

Feng & Hunter (2024) briefly consider sentence pairs where a premise $a$ entails both $b$ and the negation of $b$. They observe that in classical propositional logic, this can only hold if $a$ is contradictory in itself. They confirm that in a random sample of 1000 sentence pairs from eSNLI, each premise is non-contradictory.

### 3.2 Inference relations

Wang et al. (2019) formalize the NLI inference relations in propositional logic by:

$$E : a \rightarrow b \quad C : a \rightarrow \neg b \quad N : a \perp b$$

where $\perp$ means that there is "no clear relation between $a$ and $b$". A similar definition is given by Banerjee et al. (2024). There is a crucial oversight in both accounts: If the implication is classical, then $E$ and $C$ together exhaust the whole space of sentence pairs, and there is no room for any pair being classified as neutral. The reason is that the sentence $\neg(a \rightarrow b) \land \neg(a \rightarrow \neg b)$ is a contradiction in classical logic. So the NLI relations can not be meaningfully expressed in classical propositional logic.

Sia et al. (2023) seem to recognise that classical propositional logic will not do, in that their neutral relation is expressed as "$u \overset{?}{\Rightarrow} x$", where "$\overset{?}{\Rightarrow}$ indicates *truth-valueless*". It is unclear exactly how their application of free logic to NLI is supposed to work, as they never use the non-denoting singular terms that are the distinguishing feature of free logic. Their claim that classical logic "requires each singular term to denote a Boolean variable in the domain" suggest a mix-up of terms and formulas.

MacCartney & Manning (2014) express the three-way formulation of NLI with $a \vDash b$ for entailment, $a \vDash \neg b$ for contradiction, and $a \nvDash b \land a \nvDash \neg b$ for neutral, where $\vDash$ is some semantic notion of entailment. By contrast, Tian et al. (2021), in their study of the capabilities of language models to reason in first-order logic, use a notion of "syntactical consequence"

supposedly meaning derivability in first-order logic. They use the syntactic consequence relation $\vdash$ to define four inference relations, where $E$ is defined as $a \vdash b \wedge a \nvdash \neg b$, $C$ as $a \nvdash b \wedge a \vdash \neg b$, $N$ as $a \nvdash b \wedge a \nvdash \neg b$, plus a version of the principle of explosion that they call "Paradox": $a \vdash b \wedge a \vdash \neg b$. These definitions entail that the standard NLI relations can only hold for pairs in which the premise is non-contradictory. They also observe that the "Paradox" scenario is rare in texts as well as in NLI tasks.

### 3.3 Meta-inference relations

Minervini & Riedel (2018) postulate that certain meta-inferential relations can be deduced to hold between NLI sentence pairs when the order of the sentences are swapped. They postulate that $Eab \rightarrow \neg Cba$, where $E$ and $C$ are taken to be binary predicates of first-order logic that are axiomatized by rules of this form. The validity of this rule depends on how the relation $E$ is understood. For example, it is not compatible with the understanding of Wang et al. (2019) that $Eab$ can be defined in propositional logic as $a \rightarrow b$ and $Cab$ as $a \rightarrow \neg b$. Under this reading, the relation postulated by Minervini & Riedel (2018) holds iff $a$ is true.

Li et al. (2019) take note of five meta-inferential rules, including symmetricity of contradiction, and $Eab \wedge Cbc \rightarrow Cac$. Based on these rules, Jang et al. (2022) claim that $Eab \wedge Cac \rightarrow Cbc$ is a valid principle. But under modest assumptions on the interpretation of the NLI relations, this principle can only hold under very specific conditions that drastically restrict the relations between $a$, $b$ and $c$ (see Table 2 for details).

Srikanth & Rudinger (2025) investigate the internal consistency of NLI models by means of atomic hypothesis decomposition. They prompt an LLM to break down NLI hypotheses into "atoms", each one judged by the model to be entailed by the original hypothesis. Then, e.g., for an inference $Eab$, with $b$ decomposed into atoms $b_1, \ldots, b_n$, the inference $Eab_i$ must hold for each $i \leq n$ for a model to be internally consistent. Similar meta-inferential relations can be identified for the other NLI labels. Their results show that high accuracy on NLI pairs is not indicative of internal consistency, and that incorrectly labeled examples give rise to more inconsistencies between the original NLI pair and the generated subproblems.

## 4 The modal conception of meta-inference

We propose a reading of the NLI relations that takes inspiration from classical Tarskian semantics, and the treatment of quantification over logical structures as a modal operation.[3] The metalanguage in which we define the NLI relations is a weak classical set theory. We leave the object language, as well as the notion of structure and satisfaction unspecified. This means that we are treating the NLI sentences as structurally unanalyzed, as we are only concerned with the meta-relations between NLI relations. We are never concerned with concepts such as what makes $a$ true or false, nor the internal logical form (if any) of $a$, but only with whether, for example, $Eab$ implies in the metalanguage that $\neg Cba$.

The standard Tarskian compositional clause for implication is the basis for our modal interpretation of meta-inference. Tentatively, we define

$$Eab \text{ iff } \forall \mathcal{M}(\mathcal{M} \nvDash a \vee \mathcal{M} \vDash b).$$

This can be more succinctly and suggestively expressed in the language of propositional modal logic as $\square(a \rightarrow b)$, where $\square$ is understood as universally quantifying over accessible (possible) "worlds" or situations (Goldblatt, 1992). This is the definition of strict implication given by Lewis & Langford (1959). We remain agnostic as to which kind of worlds the modal operators quantify over, and also leave the accessibility relation unspecified. If $\vDash$ is first-order satisfaction and $\square$ quantifies over all first-order structures, then $Eab$ coincides with first-order logical consequence. If the symbols are given some other interpretation, $Eab$ may or may not coincide with the consequence relation of some other logic. Since NLI inferences allow for the use of world knowledge, it seems reasonable to allow only worlds

---

[3]This style of semantics originates with Tarski (1933/1986) and is outlined in just about any modern textbook in mathematical logic, such as Mendelson (2015).

|     | $Eab$ | $Cab$ | $Nab$ |
| --- | --- | --- | --- |
| MC | $a \rightarrow b$ | $a \rightarrow \neg b$ | $\neg(a \rightarrow b) \wedge \neg(a \rightarrow \neg b)$ |
| SC | $\Box(a \rightarrow b)$ | $\Box(a \rightarrow \neg b)$ | $\Diamond(a \wedge b) \wedge \Diamond(a \wedge \neg b)$ |
| EI | $\Diamond a \wedge \Box(a \rightarrow b)$ | $\Diamond a \wedge \Box(a \rightarrow \neg b)$ | $(\Box\neg a \vee \Diamond(a \wedge b)) \wedge (\Box\neg a \vee \Diamond(a \wedge \neg b))$ |

Table 1: Three different versions of the NLI relations

that are similar enough to the real world to be accessible. We assume that the interpretation of the modal operators are kept fixed to allow comparing different examples, and that they conform with the axioms of the minimal normal modal logic $K$ (Goldblatt, 1992).

Having observed that SNLI premises are logically possible, we also consider a version of NLI relations based on the following modified version of the Tarskian clause:

$$Eab \text{ iff } \exists\mathcal{M}(\mathcal{M} \vDash a) \wedge \forall\mathcal{M}(\mathcal{M} \nvDash a \vee \mathcal{M} \vDash b).$$

The corresponding modal expression is $\Diamond a \wedge \Box(a \rightarrow b)$, where the diamond is understood as existentially quantifying over accessible worlds.

Table 1 shows the three different readings of each of the NLI relations. The first is based on the material conditional (MC) of classical propositional logic. The second corresponds to the strict conditional (SC) of Lewis & Langford (1959). The third (EI) takes into account the fact that NLI premises are logically possible, giving the operator $\Box$ *existential import*, similar to the universal premises of Aristotle's syllogistics (Lewis & Langford, 1959, p. 62ff).

Under both modal readings, modal logic $K$ proves that, for all propositional atoms $a$ and $b$, $Eab \vee Cab \vee Nab$, so that the relations together cover the whole space.[4] For the SC reading, we can prove $\neg Eab \vee \neg Cab \vee \neg Nab$ and $Nab \rightarrow \neg Eab \wedge \neg Cab$, meaning that the relations are not fully mutually exclusive: There might be overlap between $E$ and $C$. The EI reading, on the other hand, gives full $K$-provable trichotomy:

$$(Eab \wedge \neg Cab \wedge \neg Nab) \vee (\neg Eab \wedge Cab \wedge \neg Nab) \vee (\neg Eab \wedge \neg Cab \wedge Nab).$$

Using these definitions of the NLI relations, we establish a number of meta-inferential relations. They are presented in Table 2.[5] We observe that the reading of the NLI relations with existential import stand out from the others. It validates certain principles that the others invalidate, and vice versa. Therefore it provides a handle for us to compare the principles with the relations induced by the NLI data and learned by the NLI models.

## 5   Data

To test meta-inferential relations, we require NLI data that has overlaps in the sentences appearing as the premise or hypothesis of multiple different items. We achieve this in two ways. First, we use the SNLI (Bowman et al., 2015) dataset, where each premise appears in up to three different items (one with each of the three inference relation labels). Second, we generate new NLI items with SNLI hypotheses as the premise. From SNLI and the generated items, we create a third dataset of *inferred* items which combine up to two items from the other two datasets to form a new premise–hypothesis pair. While we don't have ground-truth labels for these items, the NLI label of the original items allow us to infer (or rule out) labels depending on the logical reading of the inference relation (see Table 2).

### 5.1   SNLI

SNLI (Bowman et al., 2015) is a large dataset of human-generated NLI items, which has been a staple in the field of Computational Linguistics since its release. As premises, the

---

[4]Proofs are verified using the Tree Proof Generator (Schwarz, 2025).

[5]For brevity, we assume that the precedence of logical operators are $\neg, \wedge, \vee, \rightarrow$, so that, for example $\neg a \wedge b \vee c$ is equivalent to $((\neg a) \wedge b) \vee c)$.

|  | MC | SC | EI | Paper |
|---|---|---|---|---|
| $Eaa$ | ✓ | ✓ | $\Diamond a$ | ⋆ |
| $Caa$ | $\neg a$ | $\neg\Diamond a$ | $\bot$ | |
| $Naa$ | $\bot$ | $\bot$ | $\neg\Diamond a$ | |
| $Cab \to Cba$ | ✓ | ✓ | $\neg\Diamond a \vee \Diamond b$ | ⋆,† |
| $Cab \to \neg Eba$ | $b$ | $\Diamond b$ | ✓ | |
| $Eab \to \neg Cba$ | $a$ | $\Diamond a$ | ✓ | ⋆ |
| $Nab \to \neg Cba$ | $\varnothing$ | ✓ | $\Diamond a \vee \neg\Diamond b$ | ⋆ |
| $Eab \wedge Ebc \to Eac$ | ✓ | ✓ | ✓ | ⋆,† |
| $Eab \wedge Cbc \to Cac$ | ✓ | ✓ | ✓ | † |
| $Nab \wedge Ebc \to \neg Cac$ | $\varnothing$ | ✓ | ✓ | † |
| $Nab \wedge Cbc \to \neg Eac$ | $\varnothing$ | ✓ | ✓ | † |
| $Cab \wedge Nbc \to \neg Eca$ | $\varnothing$ | ✓ | $\alpha$ | |
| $Eab \wedge Ebc \to \neg Cca$ | $a$ | $\Diamond a$ | ✓ | |
| $Eab \wedge Cbc \to Cca$ | ✓ | ✓ | $\beta$ | |
| $Eab \wedge Cbc \to \neg Eca$ | $a \wedge \neg b \vee c$ | $\Diamond(a \wedge \neg b \vee c)$ | ✓ | |
| $Nab \wedge Ebc \to \neg Cca$ | $\varnothing$ | ✓ | $\gamma$ | |
| $Eab \wedge Cac \to \neg Ebc$ | $a \vee b \wedge \neg c$ | $\Diamond(a \vee b \wedge \neg c)$ | ✓ | |
| $Eab \wedge Cac \to Cbc$ | $a \vee \neg b \vee \neg c$ | $\delta$ | $\varepsilon$ | ‡ |
| $Eab \wedge Nac \to Nbc$ | $\varnothing$ | ✓ | ✓ | |
| $Nab \wedge Eac \to \neg Cbc$ | $\varnothing$ | ✓ | ✓ | |
| $Nab \wedge Cac \to \neg Ebc$ | $\varnothing$ | ✓ | ✓ | ‡ |

Table 2: Meta-inferential relations. We use ✓, $\bot$ and $\varnothing$ to denote, respectively, that the relation is valid, contradictory, and vacuously true. When a formula, such as $\Diamond a$, is listed, the relation is equivalent over $K$ to that formula. The formulas denoted by Greek letters are: $\alpha = \Diamond(b \vee \neg a \wedge c) \vee \Box(\neg a \vee \neg c)$, $\beta = \Box(\neg a \vee \neg b) \vee \Diamond(a \wedge \neg b \vee c)$, $\gamma = \Box(\neg b \vee \neg c) \vee \Diamond(a \vee b \wedge \neg c) \vee \Diamond(a \wedge c)$, $\delta = \Box(a \vee \neg b \vee \neg c) \vee \Diamond(a \wedge (\neg b \vee c))$ and $\varepsilon = \delta \vee \Box(\neg a \vee c)$. The paper by Minervini & Riedel (2018) is denoted ⋆, Li et al. (2019) as †, and Jang et al. (2022) as ‡.

dataset uses the visually-descriptive image captions of the Flickr30k corpus (Young et al., 2014). For each caption, crowdsourced annotators were asked to write three hypothesis sentences corresponding to each of the three NLI relations, using the instructions described in Sec. 2 of this paper. Because of this procedure, most premise sentences are repeated in three different items in the dataset. The full dataset consists of about 570k items including development and test splits of 10k each. About 10% of the data was validated by up to four additional crowd workers who independently annotated the premise–hypothesis pair with one of the NLI labels.

## 5.2 Generated NLI datasets

To generate NLI items, we provided an LLM with a prompt similar to the instructions given to the SNLI annotators. Rather than an image caption, we used the hypothesis of an SNLI item as the premise and instruct the LLM to generate a hypothesis sentence for each of the three NLI labels. In addition to the instructions, we prompted the model with 10 few-shot examples drawn from the collection of "perfect items" in the SNLI train set; that is, captions for which all four of the secondary annotators agreed with the original label on all three items.[6] The hypotheses in these examples were lightly edited to remove anaphoric references to the premise (see Appendix C for the full prompts). We repeated this procedure for 25k items sampled from the SNLI train set and all of the test set items, resulting in a dataset of 75k train items and 30k test items for each LLM.

---

[6] There are only 19 captions for which there is full inter-annotator agreement on all three corresponding SNLI items (one for each NLI label). Of these 19 we sampled 10 captions, and used the corresponding 30 SNLI items.

We experimented with datasets generated by three different LLMs: Llama3.2 (3b parameters), Llama3.3 (70b parameters), and the distilled DeepSeek-R1 model that uses the Llama3.3 architecture (70b parameters). These three models were chosen because they are open access and have quantized versions that are possible to run with commercially available cloud GPUs (see Appendix D for implementation details). For our purposes, it is important that the model can generate NLI items that aren't too far out of distribution from SNLI, which we validate in Section 6.1.

We refer to the generated datasets as Ll3.2, Ll3.3 and DS-R1, respectively. We also use combined training sets – SNLI+Ll3.2, for example, refers to the SNLI train set combined with the Llama3.2-generated train set, and SNLI±Ll3.2 is the same, but with 75k randomly sampled items from SNLI removed to maintain size parity with the original SNLI train set.

### 5.3 Inferred NLI test set

Using the SNLI and generated test sets as input, we create an inferred test set according to the following rules:

- Add $(b, a)$ for each SNLI items $(a, b, l_1)$.
- Add $(b, c)$ for each pair of SNLI items $(a, b, l_1)$ and $(a, c, l_2)$ .
- Add $(a, c)$ for each SNLI item $(a, b, l_1)$ and generated item $(b, c, l_2)$
- Add $(c, a)$ for each SNLI item $(a, b, l_1)$ and generated item $(b, c, l_2)$

While these items are presented as unlabeled here, certain values of $l_1$ and $l_2$ allow us to infer a restricted set of possible labels, depending of the meta-inferential properties of how the entailment relations are interpreted, as detailed in Table 2.

### 5.4 Manual validation

To validate the generated and inferred data, a small-scale manual annotation was carried out on 100 items. A total of 9 items (4 generated and 5 inferred) had expected labels inconsistent with the label chosen by the annotators, although the inherent subjectivity of the task can explain many of those discrepancies. The validation is described in detail in Appendix A.

## 6 Experiments

We proceed with two experiments. In Sec. 6.1 we evaluate the suitability of three LLMs for generating NLI data and select a dataset to proceed with. In Sec. 6.2 we use the inferred test set and a recently state-of-the-art NLI model to investigate the meta-inferential properties encoded in SNLI.

For each model, we also train a *hypothesis-only* baseline model, which is trained to predict the NLI label based on the hypothesis alone (Poliak et al., 2018). In the full model, the premise and hypothesis are concatenated as text input with the appropriate sentence-separator token (specific to to the base model architecture) between the two sentences. Additional training details can be found in Appendix E.

### 6.1 Suitability of the generated NLI data

We conduct the first experiment primarily with BERT (Devlin et al., 2019) since its performance on SNLI is very well-studied. We experiment with five different training sets: The standard SNLI and SNLI augmented with each of the three generated train sets, removing a randomly sampled 75k items from the SNLI train set to maintain parity with the vanilla SNLI. For Ll3.2, we also experiment with not removing the sampled items.

We observe that the vanilla SNLI model (first row of Table 3) performs about a bit worse on Ll3.2 and DS-R1 and a bit better on Ll3.3 in comparison to the SNLI test set. All models

| Model | Train set | SNLI | | Ll3.2 | | Ll3.3 | | DS-R1 | |
|---|---|---|---|---|---|---|---|---|---|
| BERT | SNLI | 77.9 | 15.0 | 73.6 | 6.1 | 81.0 | 2.1 | 73.3 | 6.4 |
| | SNLI+DS-R1 | 76.7 | 12.4 | 76.5 | 6.8 | 82.9 | 0.6 | 76.1 | 4.6 |
| | SNLI+LL3.2 | 77.6 | 14.3 | 79.3 | 8.2 | 82.0 | 1.9 | 72.9 | 5.9 |
| | SNLI+LL3.3 | 78.1 | 15.4 | 76.3 | 7.3 | 86.0 | 3.3 | 76.2 | 8.7 |
| | SNLI±DS-R1 | 78.6 | 14.7 | 77.7 | 8.0 | 84.7 | 3.0 | 78.5 | 7.2 |
| | SNLI±LL3.2 | 77.1 | 14.3 | 79.9 | 7.7 | 83.5 | 2.5 | 76.0 | 7.3 |
| | SNLI±LL3.3 | 79.1 | 16.3 | 77.0 | 7.9 | 86.9 | 4.6 | 78.0 | 8.8 |
| RoBERTa+SE | SNLI±DS-R1 | 90.7 | 20.9 | 87.5 | 17.0 | 97.4 | 10.9 | 92.4 | 15.3 |
| | SNLI±LL3.3 | 91.2 | 21.3 | 84.9 | 14.2 | 98.0 | 7.7 | 89.5 | 15.5 |

Table 3: Macro-average F1-score (and percentage point increase over the hypothesis-only model) on the standard SNLI and LLM-generated test sets. Rows correspond to different model architectures and training sets, including the standard SNLI training set and SNLI augmented with generated data, as described in §5.2.

perform well on the original SNLI data (first column of Table 3), with the LL3.3 and DS-R1-trained models providing a small boost over the unaugmented training data, even when the overall training set size was maintained.

Overall, these results suggest that the generated NLI data, especially that of the two 70b parameter models, adequately replicates the distribution of the human-generated SNLI dataset. While including the Ll3.3 data in the train set provides the largest performance boost on the SNLI test set, DS-R1 is not far behind in this regard. Moreover, we note that the difference between the performance of the full and hypothesis-only models is smaller for LL3.3 in comparison to DS-R1, suggesting that DeepSeek-R1 is less prone to "give away" the NLI label in how it generates the hypothesis in comparison to LLama3.3. For this reason, we conduct the next experiment with inferred items based on the SNLI DS-R1 test sets.

### 6.2 Meta-inferential consistency

For the main experiment, we use the inferred test set to investigate the meta-inferential properties that a model learns from SNLI. The strength of the conclusions we can draw depends somewhat on the model's NLI performance, since we assume that the model predictions follow some internal logic that is learned from the data. For this reason, the main experiment is conducted with RoBERTa+SE (Sun et al., 2020), a span-based "self-explaining" model based on RoBERTa (Zhuang et al., 2021), which has recent state-of-the-art performance on SNLI. The model is trained on SNLI±DS-R1.[7] Please refer to Table 3 for the model's performance on the SNLI and generated test sets.

We test the model on novel "inferred" items that combine sentences from standard SNLI and DS-R1 test sets, as described in Sec. 5.3. Table 4 shows results on items for which the model correctly labeled the items it was based on, since these results are most useful for assessing the model's meta-inferential consistency (see Appendix B for unfiltered results).

## 7 Discussion

We have presented two different modal readings of the NLI relations that validate different collections of meta-inferential relations, and have compared them with the labels predicted by the model. The results show that, overall, the model performs very well on the meta-inference task, as most scores are in the high 90's. As a general pattern, whenever a hypothesis $a$ is labeled with a label L in the input pattern and the same $a$ reappears as the hypothesis of the target pair, the model prediction is skewed towards the label L.

---

[7]We use the training set that includes generated data to mitigate the possibility that idiosyncrasies in the LLM-generated data would skew results on inferred items that include generated sentences.

| input items | item | $c$ | count | E | C | N | SC | SC✓ | EI | EI✓ |
|---|---|---|---|---|---|---|---|---|---|---|
| $Cab$ | | | 3023 | 0.4 | 79.5 | 20.1 | $Cba$ | 79.5 | $\neg Eba$ | 99.6 |
| $Eab$ | $ba$ | – | 3094 | 8.4 | 2.4 | 89.1 | – | – | $\neg Cba$ | 97.6 |
| $Nab$ | | | 2800 | 12.8 | 8.7 | 78.5 | $\neg Cba$ | 91.3 | – | – |
| $Eab \wedge Ebc$ | | | 2975 | 96.8 | 0.8 | 2.4 | $Eac$ | 96.8 | $Eac$ | 96.8 |
| $Eab \wedge Cbc$ | $ac$ | g | 3001 | 0.7 | 98.3 | 1.0 | $Cac$ | 98.3 | $Cac$ | 98.3 |
| $Nab \wedge Ebc$ | | | 2635 | 59.0 | 3.0 | 38.0 | $\neg Cac$ | 97.0 | $\neg Cac$ | 97.0 |
| $Nab \wedge Cbc$ | | | 2670 | 0.3 | 87.1 | 12.6 | $\neg Eac$ | 99.7 | $\neg Eac$ | 99.7 |
| $Cab \wedge Nbc$ | | | 2614 | 0.5 | 60.6 | 38.9 | $\neg Eca$ | 99.5 | – | – |
| $Eab \wedge Ebc$ | $ca$ | g | 2975 | 1.3 | 4.4 | 94.3 | – | – | $\neg Cca$ | 95.6 |
| $Eab \wedge Cbc$ | | | 3001 | 0.1 | 74.3 | 25.5 | $Cca$ | 74.3 | $\neg Eca$ | 99.9 |
| $Nab \wedge Ebc$ | | | 2635 | 2.3 | 11.7 | 86.0 | $\neg Cca$ | 88.3 | – | – |
| $Eab \wedge Cac$ | | | 2824 | 0.4 | 82.1 | 17.5 | – | – | $\neg Ebc$ | 99.6 |
| $Eab \wedge Nac$ | $bc$ | h | 2408 | 0.6 | 3.8 | 95.6 | $Nbc$ | 95.6 | $Nbc$ | 95.6 |
| $Nab \wedge Eac$ | | | 2408 | 57.2 | 6.7 | 36.1 | $\neg Cbc$ | 93.3 | $\neg Cbc$ | 93.3 |
| $Nab \wedge Cac$ | | | 2455 | 0.8 | 84.3 | 14.9 | $\neg Ebc$ | 99.2 | $\neg Ebc$ | 99.2 |

Table 4: Meta-inferential results for RoBERTa+SE trained on SNLI±DS-R1 (filtered). Results are disaggregated by the labels of the item(s) they are based on (**input items**) and the premise-hypothesis pair provided to the model (**item**). All $a$ and $b$ sentences come from SNLI — Flickr30k captions and human-generated hypotheses, respectively. The next column shows the origin of the $c$ sentence (h = SNLI hypothesis; g = LLM-generated hypothesis). The number of items included is listed under **count**; **E**, **C** and **N** give the percentage of model predictions for each label; **SC** shows what we can infer (if anything) about the expected label under the strict conditional reading and **SC✓** is the percentage of predictions consistent with the possible label set implied by the labels of the input items under that reading (see table 2); similarly for columns **EI** and **EI✓** under the existential import reading.

If, for a given relation, the labels inferred under one of the modal readings agree with the model prediction, we take that as evidence for that reading being compatible with the prediction of the model. If one of the readings exhibit stronger agreement, this suggests that that reading is more compatible with the model's notion of inference. We discuss the following three items: (1) inferences where the SC and the EI readings predict different labels; (2) inferences scoring below 96; and (3) inferences that score surprisingly high.

First, consider the meta-inference $Cab \rightarrow Cba$, which is valid under SC, but not under EI. The SC reading agrees with the model prediction in 79.5% of of the cases. By contrast, the meta-inference $Cab \rightarrow \neg Eba$ is invalid under SC, but valid under EI. The EI reading agrees with the model in 99.6% of the cases. Similarly, the meta-inference $Eab \wedge Cbc \rightarrow \neg Eca$, which is valid under EI (model scores 99.9), but where SC instead predicts $Cca$ (score 74.3). This initially suggests that the EI reading of the NLI relations is more compatible with the model predictions.

We manually analyze 100 randomly sampled items corresponding to the meta-inferences $Cab \rightarrow Cba/\neg Eba$ discussed in the previous paragraph. The first group draws 50 items for which both $ab$ and $ba$ were labeled $C$, and the second 50 items where $ab$ were labeled $C$ but $ba$ were labeled $N$ – which is correct under the EI reading, but not under SC. The second group contains 25 items exhibiting "backwards anaphora" ("The man ..." in premise and "A man ..." in hypothesis). There are 7 items where both $ab$ and $ba$ can be construed as neutral. Three premises contain an explicit "not", and one premise is in the imperative. By contrast, the first group contain no mislabelled items, only 8 examples of backwards anaphora, and none of the other artifacts. This suggests that artifacts in the data may contribute to the relatively high number of items being labeled $N$ for this meta-inference.

There are five other meta-inferences where the model scores less than 96. The lowest score, 88.3, is for the meta-inference $Nab \wedge Ebc \rightarrow \neg Cca$. Interestingly, this pattern is not valid on the EI reading, but only under the MC (and vacuously so) and SC readings. The second

lowest score, 91.3 is for the pattern $Nab \rightarrow \neg Cba$, which is also invalid under the EI reading. Since it is not possible to infer a non-trivial set of possible labels for these inferences under the EI reading, there is no clear-cut comparison to be made. By contrast, the meta-inference $Eab \wedge Ebc \rightarrow \neg Cca$ scores the relatively low 95.6, but is only valid under the EI reading.

The remaining two low-scoring meta-inferences are $Eab \wedge Nac \rightarrow Nbc$, scoring 95.6, and $Nab \wedge Eac \rightarrow \neg Cbc$, scoring 93.3. They are both valid under the EI reading. Note that the first of these meta-inferences is equivalent over $K$ to $Nab \wedge Eac \rightarrow Ncb$, which together with the meta-inference $Nab \rightarrow \neg Cba$ would make it possible to deduce the latter meta-inference. But since $Nab \rightarrow \neg Cba$ is invalid on the EI reading, this deduction is not possible. This seems to suggest that the EI reading is more compatible with the model predictions. On the other hand, the inference $Cab \wedge Nbc \rightarrow \neg Eca$ scores 99.5, even though it is invalid under EI.

All but one of the meta-inferences for which the model scores high are either valid under both readings, or valid only under EI. Two of the lower scores occur in connection to the meta-inference being invalid under either the SC or EI reading. The remaing two lower scores are at least in part related to a meta-inference that is invalid under EI. So in case there is a difference between the two readings, validity under EI is correlated with a high degree of agreement with the model prediction. Note that there is only one meta-inference that is valid only on the SC reading that scores higher than the EI reading.

All in all, this suggests that EI is indeed more compatible with the model prediction than SC is. It also suggests that it is not yet possible to completely disentangle the contributions of the different readings, since invalidity under any of the readings correlates with low scores.

## 8 Future work

One of the long-standing problems in NLI research is the type of inference relations that are captured in different NLI datasets. It is well known that different NLI datasets capture different aspects of inference, e.g., compare the stricter logical entailment relations as these are encoded in the FraCaS test suite to the more common-sense inference type relations as these are found in the RTE datasets. We believe that our method can potentially lead to a principled disentaglement of these relations on the basis of solid logical underpinnnings. We aspire to provide an attempt at this disentaglement via logics in our future work. For example, the guidelines for RTE-5 (TAC2009, 2009) force hypotheses to be possibly false, which gives rise to yet another different modal notion of entailment.

We would like to further examine the implicit closed-world assumption of the SNLI premises – that they describe a situation exhaustively enough. This is likely to affect which hypotheses would be considered contradictions, and how those are constructed by human workers, as well as having bearing on how to resolve coreference on premise–hypothesis pairs. Relatedly, we would like to conduct a more in-depth investigation of possible discourse effects between premise and hypothesis. This study assumes that the same sentence always expresses the same proposition and, while we took steps to reduce the use of anaphora in generated hypotheses, we know that the original SNLI data does contain anaphora.

Another more general avenue that we want to pursue is the use of logics to probe/under-stand/approximate better what modern NLU models learn from a given task. We want to argue that whatever your stance on the use of logics for NLP is, they are at least useful as a generic, robust framework for probing into the reasoning capabilities of these models and as such serious engagement with various well-founded logic systems is helpful in making sense of these capabilities.

## Acknowledgments

This work was supported by grant 2014-39 from the Swedish Research Council (VR) for the establishment of the Centre for Linguistic Theory and Studies in Probability (CLASP) at the University of Gothenburg. Stergios Chatzikyriakidis gratefully acknowledges funding from Amazon/GRNET for the project Neural-Symbolic Integration for Enhanced Natural Language Processing (NIELS).

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

| | generated | $ab; ba$ | $ab, ac; bc$ | $ab, bc; ac$ | $ab, bc; ca$ |
|---|---|---|---|---|---|
| annotator disagreements | 3 | 2 | 1 | 3 | 5 |
| substantial disagreements (EI) | 0 | 2 | 0 | 0 | - |
| substantial disagreements (SC) | 2 | 2 | 0 | 0 | - |
| inconsistent consensus label | 4 | 0 | 0 | 5 | 0 |
| total annotated items | 20 | 20 | 20 | 20 | 20 |

Table 5: Counts for the annotated items by category. The *substantial disagreements* rows show disagreements where at least one of the annotated labels was *not* among the item's inferred possible labels under the respective reading. The *inconsistent consensus label* row shows counts of items for which the annotators' consensus label was outside of the possible label set for the inferred or generated item.

# A  Manual validation

In order to validate the generated and inferred items, a small-scale manual annotation was carried out by two of the authors. Both annotators assigned an NLI label to each of 100 items – 20 items generated by Llama3.3 and 20 from each of the four inference patters described in Section 5.3. The annotators then discussed disagreements and decided on a consensus label. The items were shuffled prior to annotation, and the source and expected labels were unknown at the time of annotation.

Results of the annotation can be found in Table 5. The agreement between the annotators was good overall. A total of 14 items required discussion to reach a consensus. Of the disagreements, several represented inferred items where both of the annotated labels were in set of possible labels. I.e., they were not *substantial disagreements* under one or both of the meta-inferential readings.

We found a total of 9 (4 generated and 5 from the inference pattern $ab, bc; ac$) where the consensus annotated label is not consistent with the expected label set (either the label used to prompt the model or the inferred possible labels). Table 6 shows those items, the expected labels, and how they were annotated. We note that in two of the four generated items, one of the annotators initially chose the label intended by the model. In nearly all cases, the inconsistency appears to hinge on either the precise interpretation of a lexical item or degree of plausibility of the hypothesis in conjunction with the premise. This observation highlights the subjective nature of the task, but overall gives us confidence that the data is of similar enough quality to SNLI for the present purposes.

# B  Additional results

This section contains additional results for the experiment in Sec. 6.2. In particular, Table 7 shows the results *without* filtering out items for which the model made an incorrect prediction on one or more of the anticedent input items; and Tables 8 and 9 show the same results for the Llama3.3-generated data.

As in Table 4: Results are disaggregated by the labels of the item(s) they are based on (**input items**) and the premise-hypothesis pair provided to the model (**item**). All $a$ and $b$ sentences come from SNLI — Flickr30k captions and human-generated hypotheses, respectively. The next column shows the origin of the $c$ sentence (h = SNLI hypothesis; g = LLM-generated hypothesis). The number of items included is listed under **count**; **E**, **C** and **N** give the percentage of model predictions for each label; **SC** shows what we can infer (if anything) about the expected label under the strict conditional reading and **SC✓** is the percentage of predictions consistent with the possible label set implied by the labels of the input items under that reading (see table 2); similarly for columns **EI** and **EI✓** under the existential import reading.

| source | item | Ex | Cn | A1 | A2 |
|---|---|---|---|---|---|
| generated | P: A man is teaching his granddaughter to swim.
H: A grandfather is instructing a girl on swimming techniques. | N | E | E | E |
| | P: People near a lot of reading materials.
H: People are playing video games | C | N | N | N |
| | P: The track runners are all black ethniticity
H: The runners are people of African descent | E | N | E | N |
| | P: they are singing in a christmas program
H: People are performing on stage. | E | N | E | N |
| $ab, bc; ac$ | P: A man in a green shirt is getting his feet wet in front of a wall showing a closeup picture of a face.
H: There's a man outdoors. | E | N | N | N |
| | P: Three people posing for a camera
H: All individuals are completely still. | C,N | E | E | E |
| | P: A lady in pink long-sleeved blouse is holding a pink bag in her lap as she reaches something to the guy next to her.
H: a woman in pink shakes hands with a man | E,N | C | C | C |
| | P: A group of young boys wearing track jackets stretch their legs on a gym floor as they sit in a circle.
H: The group is sitting down quietly. | C | N | N | N |
| | One man, shirtless, and a woman, clothed, jumping in the water.
They are completely dry. | C | N | N | N |

Table 6: Error analysis for generated and inferred items. Column **Ex** shows the *expected labels*, which is either the label Llama3.3 was prompted to generate or the possible labels inferred from the labels of the original items. Note that the five $ab, bc; ac$ items had the same inferred labels under both the SC and EI readings. **Cn** shows the consensus label, and **A1** and **A2** show the labels selected by the two annotators. This table shows all annotated items for which the consensus label was not among the expected labels (i.e., where **Cn** does not appear in **Ex**).

| input items | item | c | count | E | C | N | SC | SC✓ | EI | EI✓ |
|---|---|---|---|---|---|---|---|---|---|---|
| *Cab* | | – | 3237 | 0.9 | 76.2 | 22.9 | *Cba* | 76.2 | ¬*Eba* | 99.1 |
| *Eab* | *ba* | | 3368 | 8.7 | 3.1 | 88.2 | – | – | ¬*Cba* | 96.9 |
| *Nab* | | | 3219 | 12.0 | 10.0 | 78.0 | ¬*Cba* | 90.0 | – | – |
| *Eab* ∧ *Ebc* | | | 3368 | 93.0 | 1.7 | 5.3 | *Eac* | 93.0 | *Eac* | 93.0 |
| *Eab* ∧ *Cbc* | *ac* | g | 3368 | 1.1 | 96.3 | 2.6 | *Cac* | 96.3 | *Cac* | 96.3 |
| *Nab* ∧ *Ebc* | | | 3219 | 58.2 | 4.5 | 37.4 | ¬*Cac* | 95.5 | ¬*Cac* | 95.5 |
| *Nab* ∧ *Cbc* | | | 3219 | 0.7 | 85.3 | 13.9 | ¬*Eac* | 99.3 | ¬*Eac* | 99.3 |
| *Cab* ∧ *Nbc* | | | 3237 | 0.5 | 59.3 | 40.3 | ¬*Eca* | 99.5 | – | – |
| *Eab* ∧ *Ebc* | *ca* | g | 3368 | 1.2 | 5.3 | 93.4 | – | – | ¬*Cca* | 94.7 |
| *Eab* ∧ *Cbc* | | | 3368 | 0.1 | 73.5 | 26.4 | *Cca* | 73.5 | ¬*Eca* | 99.9 |
| *Nab* ∧ *Ebc* | | | 3219 | 2.1 | 12.6 | 85.3 | ¬*Cca* | 87.4 | – | – |
| *Eab* ∧ *Cac* | | | 3247 | 0.7 | 78.9 | 20.3 | – | – | ¬*Ebc* | 99.3 |
| *Eab* ∧ *Nac* | *bc* | h | 2858 | 2.7 | 6.6 | 90.7 | *Nbc* | 90.7 | *Nbc* | 90.7 |
| *Nab* ∧ *Eac* | | | 2858 | 53.1 | 8.2 | 38.6 | ¬*Cbc* | 91.8 | ¬*Cbc* | 91.8 |
| *Nab* ∧ *Cac* | | | 2951 | 1.5 | 81.4 | 17.1 | ¬*Ebc* | 98.5 | ¬*Ebc* | 98.5 |

Table 7: Meta-inferential results for RoBERTa+SE trained on SNLI±DS-R1 (unfiltered).

| input items | item | c | count | E | C | N | SC | SC✓ | EI | EI✓ |
|---|---|---|---|---|---|---|---|---|---|---|
| *Cab* | | – | 3029 | 0.4 | 87.3 | 12.3 | *Cba* | 87.3 | ¬*Eba* | 99.6 |
| *Eab* | *ba* | | 3096 | 7.7 | 3.3 | 89.0 | – | – | ¬*Cba* | 96.7 |
| *Nab* | | | 2834 | 12.0 | 9.3 | 78.7 | ¬*Cba* | 90.7 | – | – |
| *Eab* ∧ *Ebc* | | | 2953 | 96.9 | 0.8 | 2.3 | *Eac* | 96.9 | *Eac* | 96.9 |
| *Eab* ∧ *Cbc* | *ac* | g | 2975 | 1.3 | 97.1 | 1.5 | *Cac* | 97.1 | *Cac* | 97.1 |
| *Nab* ∧ *Ebc* | | | 2678 | 58.6 | 3.3 | 38.1 | ¬*Cac* | 96.7 | ¬*Cac* | 96.7 |
| *Nab* ∧ *Cbc* | | | 2680 | 0.6 | 82.8 | 16.6 | ¬*Eac* | 99.4 | ¬*Eac* | 99.4 |
| *Cab* ∧ *Nbc* | | | 2300 | 0.6 | 67.8 | 31.7 | ¬*Eca* | 99.4 | – | – |
| *Eab* ∧ *Ebc* | *ca* | g | 2953 | 1.2 | 7.0 | 91.8 | – | – | ¬*Cca* | 93.0 |
| *Eab* ∧ *Cbc* | | | 2975 | 0.1 | 82.4 | 17.5 | *Cca* | 82.4 | ¬*Eca* | 99.9 |
| *Nab* ∧ *Ebc* | | | 2678 | 2.5 | 14.1 | 83.4 | ¬*Cca* | 85.9 | – | – |
| *Eab* ∧ *Cac* | | | 2834 | 0.5 | 82.9 | 16.6 | – | – | ¬*Ebc* | 99.5 |
| *Eab* ∧ *Nac* | *bc* | h | 2439 | 0.6 | 3.9 | 95.5 | *Nbc* | 95.5 | *Nbc* | 95.5 |
| *Nab* ∧ *Eac* | | | 2439 | 55.7 | 7.5 | 36.8 | ¬*Cbc* | 92.5 | ¬*Cbc* | 92.5 |
| *Nab* ∧ *Cac* | | | 2497 | 1.1 | 82.3 | 16.6 | ¬*Ebc* | 98.9 | ¬*Ebc* | 98.9 |

Table 8: Meta-inferential results for RoBERTa+SE trained on SNLI±Ll3.3 (filtered).

| input items | item | $c$ | count | E | C | N | SC | SC✓ | EI | EI✓ |
|---|---|---|---|---|---|---|---|---|---|---|
| $Cab$ | | | 3237 | 0.8 | 83.7 | 15.5 | $Cba$ | 83.7 | $\neg Eba$ | 99.2 |
| $Eab$ | $ba$ | − | 3368 | 8.2 | 4.0 | 87.7 | − | − | $\neg Cba$ | 96.0 |
| $Nab$ | | | 3219 | 11.3 | 10.9 | 77.7 | $\neg Cba$ | 89.1 | − | − |
| $Eab \wedge Ebc$ | | | 3368 | 92.6 | 1.6 | 5.7 | $Eac$ | 92.6 | $Eac$ | 92.6 |
| $Eab \wedge Cbc$ | $ac$ | g | 3368 | 1.7 | 94.7 | 3.6 | $Cac$ | 94.7 | $Cac$ | 94.7 |
| $Nab \wedge Ebc$ | | | 3219 | 58.2 | 4.6 | 37.2 | $\neg Cac$ | 95.4 | $\neg Cac$ | 95.4 |
| $Nab \wedge Cbc$ | | | 3219 | 1.1 | 81.8 | 17.1 | $\neg Eac$ | 98.9 | $\neg Eac$ | 98.9 |
| $Cab \wedge Nbc$ | | | 3237 | 0.6 | 65.5 | 33.9 | $\neg Eca$ | 99.4 | − | − |
| $Eab \wedge Ebc$ | $ca$ | g | 3368 | 1.2 | 7.7 | 91.1 | − | − | $\neg Cca$ | 92.3 |
| $Eab \wedge Cbc$ | | | 3368 | 0.1 | 80.9 | 18.9 | $Cca$ | 80.9 | $\neg Eca$ | 99.9 |
| $Nab \wedge Ebc$ | | | 3219 | 2.3 | 15.2 | 82.5 | $\neg Cca$ | 84.8 | − | − |
| $Eab \wedge Cac$ | | | 3247 | 0.9 | 79.8 | 19.3 | − | − | $\neg Ebc$ | 99.1 |
| $Eab \wedge Nac$ | $bc$ | h | 2858 | 2.9 | 6.1 | 91.0 | $Nbc$ | 91.0 | $Nbc$ | 91.0 |
| $Nab \wedge Eac$ | | | 2858 | 52.6 | 8.7 | 38.7 | $\neg Cbc$ | 91.3 | $\neg Cbc$ | 91.3 |
| $Nab \wedge Cac$ | | | 2951 | 1.3 | 80.9 | 17.9 | $\neg Ebc$ | 98.7 | $\neg Ebc$ | 98.7 |

Table 9: Meta-inferential results for RoBERTa+SE trained on SNLI±Ll3.3 (unfiltered).

## C   LLM prompts for generating NLI data

This section contains the full text of the prompt provided to the LLMs (also available in the code repository). Here, we display system:, user: and assistant: tags for legibility, but the exact formatting of the prompt text depends on the language-model specific template. The {{premise}} tag was filled in with the supplied premise and was the only aspect of the prompt that varried between items.

```
system: You are native speaker of English and an expert annotator of various syntactic
    and semantic linguistic phenomena.

You have been asked to help create a Natural Language Inference (NLI) dataset. Given a
    caption of a photo, your task is to write an alternate caption that relates to the
    original caption according to each of the three NLI Labels ("entailment", "
    contradiction", and "neutral").

We will show you the caption for a photo. We will not show you the photo. Using only the
    caption and what you know about the world:

1. [entailment]: Write one alternate caption that is definitely a true description of
    the photo.
  - Example: For the caption "Two dogs are running through a field." you could write "
      There are animals outdoors."

2. [neutral]: Write one alternate caption that might be a true description of the photo.
  - Example: For the caption "Two dogs are running through a field." you could write "
      Some puppies are running to catch a stick."

3. [contradiction]: Write one alternate caption that is definitely a false description
    of the photo.
  - Example: For the caption "Two dogs are running through a field." you could write "
      The pets are sitting on a couchl"
  - This is different from the maybe correct category because it's impossible for the
      dogs to be both running and sitting.

When you get the caption from the user, please format your response as a JSON object as
    follows:

{
```

```
    "contradiction": "your 1st caption",
    "entailment": "your 2nd caption",
    "neutral": "your 3rd caption"
}

user: Premise: A smiling man holds a small child wearing pink and red.
assistant:
{
    "contradiction": "A man is hugging his dog.",
    "entailment": "A man is smiling.",
    "neutral": "A man is holding his daughter."
}
user: Premise: Two teams, one in pink and one in white, play lacrosse on a field
.
assistant:
{
    "contradiction": "Swimmers race to the finish.",
    "entailment": "Teams compete outdoors.",
    "neutral": "The pink team is winning."
}
user: Premise: Asian boy in orange, white, and black coat celebrating St. Patric
k's day with a soda and green balloon.
assistant:
{
    "contradiction": "A boy dressed in all black is hanging from a jungle gym.",
    "entailment": "a young child holds a sode and balloon for st. pats day",
    "neutral": "a young christian celebrates the holidays"
}
user: Premise: An Indian dancer or actress is performing on stage in the spotlig
ht, seated on the floor in a dramatic pose and wearing a blue dress.
assistant:
{
    "contradiction": "Jugglers performing for a group of children.",
    "entailment": "A performer sitting under a spotlight.",
    "neutral": "An actress or dancer are being critiqued for their audition."
}
user: Premise: A brown dog is running across the snowy ground.
assistant:
{
    "contradiction": "A dog is asleep.",
    "entailment": "A dog is outside.",
    "neutral": "A lab is in the snow."
}
user: Premise: A older lady is working with a mix master while two young boys st
and next to her with aprons on helping her.
assistant:
{
    "contradiction": "A lady and two boy eating waffles at their kitchen table."
,
    "entailment": "A lady mixing something.",
    "neutral": "A lady teaching her grandchildren how to bake a cake."
}
user: Premise: Young female examining a specimen in a microscope.
assistant:
{
    "contradiction": "A young female is asleep",
    "entailment": "A young female is looking into a microscope",
    "neutral": "A young female is looking at a blood sample"
}
user: Premise: A girl in a ethereal, blue costume holding a decorated hula hoop
gazes intently into the camera.
assistant:
{
    "contradiction": "a girl is dead",
    "entailment": "a girl in a costume",
```

```
    "neutral": "a girl is five"
}
user: Premise: two women sitting at a table, one looking down and the other on a
 laptop, with a banner that says sweden in spanish behind them.
assistant:
{
    "contradiction": "two people are staring directly at each other.",
    "entailment": "A couple of women are at a table.",
    "neutral": "Two women are part of a language club in their college."
}
user: Premise: A person wearing a bright pink wig and a yellow shirt, facing awa
y and looking at the ocean.
assistant:
{
    "contradiction": "A person wearing a shirt that is red.",
    "entailment": "A person is standing beside the ocean.",
    "neutral": "A person stares at the ocean as they contemplate the drastic dec
isions they just recently had to make."
}
user: Premise: {{premise}}
~
```

## D  Generative LLM details

All three LLMs were run using the Ollama API (version 0.5.10). Model details are as follows:

|  | architecture | parameters | quantization |
|---|---|---|---|
| Llama3.2 | Llama | 3.21B | Q4_K_M |
| Llama3.3 | Llama | 70.6B | Q4_K_M |
| DeepSeek-R1 | Llama | 70.6B | Q4_K_M |

## E  NLI model training details

The BERT models were trained with a batch size of 64 using the Adam optimizer (Kingma et al., 2015) with a learning rate of 0.001, $\beta_1$ =0.9 and $\beta_2$ =0.999, $\epsilon$=1e-08, and zero weight decay (default PyTorch settings). The models were given simple linear classification head for the NLI task, which took the sequence [CLS] token as input. Only the final two layers of the BERt base were trained while the first 10 layers were kept frozen.

The RoBERTa+SE models were trained with a batch size of 32. Following Sun et al. (2020), we used the AdamW, optimizer (Loshchilov & Hutter, 2019) with a learning rate of 2e-5, $\beta_1$ =0.9, $\beta_2$=0.98, $\epsilon$=10e-8, and weight decay of 0.01 (except for on bias and layer norm parameters which received no weight decay).

All models were trained for 10 epochs and the checkpoint of the best validation epoch (based on the vanilla SNLI dev set) was used for testing.

