# OpenReview forum: "Reverse-engineering NLI: A study of the meta-inferential properties of Natural Language Inference"
_colmweb.org/COLM/2025/Conference — COLM 2025_

### Official Review · Reviewer_ndSY · 2025-05-11

**Rating:** 7
**Confidence:** 4
**Ethics Flag:** 1

**Summary:**

This paper investigates how NLI models interpret inference relations and whether these interpretations align with formal logical notions. The authors propose three logical readings of NLI labels (material conditionals (MC), strict conditionals (SC), and existential import (EI)) and examine which best aligns with model behavior. They also curate datasets using the SNLI corpus, newly generated NLI items from large language models (LLaMA3.2, LLaMA3.3, and DeepSeek-R1), and an inferred test set created by composing premise–hypothesis chains from the previous datasets. NLI models (BERT and RoBERTa+SE) are evaluated after being trained on various combinations of these datasets. Results show that models perform well on meta-inference tasks overall, with RoBERTa+SE trained on SNLI and DeepSeek-R1 data showing the highest alignment. The findings support the EI reading as the most compatible with the behavior of current NLI models.

**Questions To Authors:**

You use LLMs to generate new NLI examples. How do you make sure these examples follow the same reasoning or assumptions as the original human-written ones? Did you do any quality checks or human evaluation to confirm this? It would have been helpful to include more details on this in the paper.

The paper introduces three readings of NLI relations (MC, SC, and EI), but in the discussion, it only refers to two modal readings. I understand that the MC reading is set aside, but this could be confusing for readers. Could you clarify why only two readings are discussed at that point?

**Reasons To Accept:**

The paper provides clear and well-supported explanations for the problem. It offers a much-needed critical analysis of widely used NLI datasets such as SNLI and discusses its logical inconsistencies that have been overlooked in prior work.

The paper clearly explains and compares three ways to understand NLI labels which makes the analysis more solid and connects the results to formal logic.

The use of examples generated by LLMs and the construction of inferred test sets based on compositional relations is novel.

The authors test model performance on different combinations of datasets, which helps show how data sources affect the models’ reasoning. This also creates useful directions for future research to build on.

Since NLI continues to be a benchmark task in NLP, the paper’s focus on inference about inference is relevant for understanding what NLU models are actually learning.

**Reasons To Reject:**

The study is an analysis of existing  models.

---

> ### Author Response · Authors · 2025-05-30
>
> Thank you very much for the thorough review. Please let us know if you have any questions for us – it would seem that the "questions" section currently repeats the reasons to accept.

---

> > ### Comment · Reviewer_ndSY · 2025-06-05
> >
> > I updated my questions since there was a copy/paste issue in my review.

---

> > > ### Author Response · Authors · 2025-06-05
> > >
> > > Thank you for your reply. However, we believe there may be an administrative error, as your comments refer to a "DTO method" and math problems, while our paper focuses on Natural Language Inference (NLI) and meta-inferential properties.
> > >
> > > While we do use LLMs in our work (including but not limited to LLAMA models - we also employ DeepSeek-R1), they are used for generating synthetic NLI data, not for mathematical reasoning.
> > >
> > > We suspect these comments were intended for a different paper. Could you please verify and provide feedback relevant to our NLI research?
> > >
> > > We also noticed a score change from 7 to 4 and would appreciate clarification if this was related to the mismatched comments.
> > >
> > > Thank you for your understanding.

---

> > > > ### Comment · Reviewer_ndSY · 2025-06-05
> > > > **updated questions**
> > > >
> > > > Sorry, I updated the questions. I did not mean to lower the score, I changed it back.

---

> > > > > ### Author Response · Authors · 2025-06-06
> > > > >
> > > > > No problem -- and thank you for your questions. Answers below:
> > > > >
> > > > > > You use LLMs to generate new NLI examples. How do you make sure these examples follow the same reasoning or assumptions as the original human-written ones? Did you do any quality checks or human evaluation to confirm this? It would have been helpful to include more details on this in the paper.
> > > > >
> > > > > Beyond informal sanity checking we didn't do a systematic human evaluation of the generated NLI examples.
> > > > >
> > > > > What we observed is that 1) the BERT model trained on SNLI performs well on the generated test set, suggesting that the generated examples are within-distribution for SNLI, and 2) that the BERT model trained on SNLI+generated (especially in the condition where the generated items _replace_ original SNLI items to maintain the original train set size) has slightly _better_ performance on the SNLI test set than the model trained on SNLI train alone. This gave us confidence that the generated train items are not merely trivially correct.
> > > > >
> > > > > > The paper introduces three readings of NLI relations (MC, SC, and EI), but in the discussion, it only refers to two modal readings. I understand that the MC reading is set aside, but this could be confusing for readers. Could you clarify why only two readings are discussed at that point?
> > > > >
> > > > > Thank you for pointing this out. Indeed, the MC reading is set aside in favor of modal readings since it is degenerate in the sense that there is no way to express _Neutral_. The sentence on 91 is misleading in context since we only consider two _modal_ readings. We will correct this and clarify the reason for setting MC aside in the final revision.

---

> > > > > > ### Comment · Reviewer_ndSY · 2025-06-07
> > > > > >
> > > > > > Thank you for the answers.

---

### Official Review · Reviewer_NiGL · 2025-05-13

**Rating:** 7
**Confidence:** 3
**Ethics Flag:** 1

**Summary:**

This paper revisits the interpretation of Natural Language Inference (NLI), a widely used task for evaluating language models' understanding of inference. While NLI plays a role in natural language understanding benchmarks, the logical properties of the task are not always clearly defined. The authors outline three possible interpretations of the NLI label set (material conditional, strict conditional and existential import) and examine their meta-inferential implications. They offer an extensive review on various meta-inferential inferences and test their proposed interpretations, and point out a gap in the existing literature: the “NLI relations are left undefined so that it is not clear what the relations are or in which logic they can be expressed.” However, the authors stop short of analyzing these relations in terms of standard relational properties such as transitivity, reflexivity, or symmetry (though an example in the introduction alludes to the symmetry of contradiction).  Using the SNLI dataset, they analyze examples with shared premises and supplement this with data generated by language models to evaluate the consistency of model behavior.  The findings offer a view of how different interpretations of inference may be reflected in NLI data and model responses. The discussion of future work offers directions for refining the task and enhancing its interpretability.

**Questions To Authors:**

See above

**Reasons To Accept:**

This is a well-written paper that offers an extensive theoretical review, effectively supporting the identified gaps in the existing literature. The generated data tests the introduced premises.

**Reasons To Reject:**

While this may not be a reason for rejection, one limitation of the paper is the lack of discussion regarding the choice of language models. The authors do not provide a rationale for why these specific models were selected for the tasks, or why they might be particularly suitable for assessing meta-inferential properties. It would be helpful to know whether similar behavior would be expected from other models, especially given the generality of the claims.

---

> ### Author Response · Authors · 2025-05-30
>
> Thank you for your thoughtful review.
>
> We chose the three generative models largely because they are open access (for reproducibility) and relatively affordable to run. For our purposes, it was only important that the model could generate NLI items that aren't too far out of distribution from SNLI. The analysis in §6.1 (Table 3) assured us that that was the case so we didn't turn to more costly options.
>
> For the NLI models, the strength of the conclusions we can draw from the models' predictions on the `inferred` dataset depends somewhat on the model's NLI performance since it assumes the models predictions are "correct" by some internal logic. For this reason, we use the SotA RoBERTa+SE model in §6.2. We included BERT as a strong baseline in §6.1 since it has been extensively studied in the context of NLI.
>
> We will try to clarify these points in the final version of the paper.

---

> > ### Comment · Reviewer_NiGL · 2025-06-09
> >
> > Thank you for your response!

---

### Official Review · Reviewer_yGJG · 2025-05-13

**Rating:** 6
**Confidence:** 3
**Ethics Flag:** 1

**Summary:**

The paper gives the first side-by-side formal semantics for the three SNLI labels: (i) material conditional, (ii) strict conditional, and (iii) existential import strict conditional and turns those definitions into a meta-inference test suite that measures a model’s logical self-consistency.

**Questions To Authors:**

- Can the LLM that was used to generate data affect the results if its own heuristics align with the target logic?

**Reasons To Accept:**

- Interesting formalization of the three SNLI labels under three competing logics. Mapping entailment/contradiction/neutral to three explicit logics reveals which meta-rules (symmetry, transitivity, trichotomy, etc.) are valid under each reading
- Evidence that SNLI and models trained on it are implicitly adopting an existential-import modal semantics rather than classical or pure strict implication

**Reasons To Reject:**

- The work focuses on SNLI, whose artifacts and shallow heuristics are well documented. Analysis on other datasets (e.g., MNLI) would have been beneficial. Similarly, including zero-shot results with contemporary LLMs would have strengthen the claims
- This work seems better aligned with *CL or TACL that emphasize formal semantics

---

> ### Author Response · Authors · 2025-05-30
>
> Than you for your thoughtful review. Please find our responses below.
>
> ### The work focuses on SNLI, whose artifacts and shallow heuristics are well documented. Analysis on other datasets (e.g., MNLI) would have been beneficial. Similarly, including zero-shot results with contemporary LLMs would have strengthen the claims
>
> While we agree that this analysis can and should be carried out on other datasets, the problem of artifacts and shallow heuristics is pervasive in NLI and persists in MNLI in particular (see [Gururangan et al., 2018](https://aclanthology.org/N18-2017.pdf)). This method could be adapted to probe generative LLMs for their notion of entailment, but the focus in this paper was on what is learned from the data and for this we need a high-performing fine-tuned NLI model.
>
> ### This work seems better aligned with *CL or TACL that emphasize formal semantics
>
> The model is motivated by formal semantics (in particular the connection between NLI and natural language semantics), but our core contribution is a methodology to analyze what is learned from an NLI dataset by probing a fine-tuned model for the notion of entailment it encodes. We believe this falls squarely within the scope of the COLM CfP, especially:
>
> > - All about **data**: pre-training data, alignment data, and synthetic data --- via manual or algorithmic analysis, curation, and generation
> > - All about **evaluation**: benchmarks, simulation environments, scalable oversight, evaluation protocols and metrics, human and/or machine evaluation
>
> ### Can the LLM that was used to generate data affect the results if its own heuristics align with the target logic?
>
> This is a very good question. In general, this would not affect the results, though it is conceivable that certain outputs would only be possible from an LLM following a certain logic.
>
> First for the general case: Generated sentences are used only as the premises/hypotheses $C$ on rows 8–16 of Table 2. At the point of generation, the LLM is only shown a sentence $B$ (that occurs in some SNLI sentence pair $(A,B)$) and is asked to generate a sentence $C$ for each of the three NLI labels. The model is, however, not shown the pair $(A,B)$ nor its label. Therefore, the generation of $C$ can not in any way be restricted by what relation $A$ may have to $B$ or $C$ – and it is precisely the relation between $A$ and $C$ that we are testing later on.
>
> But: For example, a model faithfully following the MC logic could, when asked to generate a contradiction to a sentence $A$, conceivably generate the very same sentence $A$, since for false $A$'s the pair $(A,A)$ is a contradiction in MC. On the other hand, a model faithfully following EI could not do so, since no pair $(A,A)$ is a contradiction on the EI reading, as shown on row 2 of Table 2.

---

> > ### Comment · Reviewer_yGJG · 2025-06-07
> > **Thank you!**
> >
> > Thank you for your response, I updated my score: $5 \rightarrow 6$

---

### Official Review · Reviewer_UYbC · 2025-05-21

**Rating:** 2
**Confidence:** 4
**Ethics Flag:** 1

**Summary:**

The paper aims to further understand logical properties in the NLI task. To do so, it uses LLMs to generate examples from SNLI, specifically using hypotheses from SNLI as premises and asking LLMs to generate new NLI hypotheses from existing hypotheses. The paper then propagates NLI labels from the existing examples to the new examples.

**Questions To Authors:**

Why is there no RoBERTa baseline trained on just SNLI in Table 3?

How exactly does the performance in Table 3 demonstrate that the "generated NLI data adequately replicates the distribution of the human-generated SNLI dataset"?

How many  “perfect items” in the SNLI train set are there (line 218)?

In 6.1, how can we conclude that BERT "performs about a bit worse on Ll3.2 and DS-R1 and a bit better on Ll3.3 in comparison to the SNLI test set” if the data isnt annotated? Are the labels just propogated based on 234-237? If so, what verification/manual sampling was done to verify these rules for propogating labels.

In Table 3, in each row, the E, C, N values sum to one. But what are we comparing to? Are about 1/3 of the examples supposed to be E, C, N?

What is gained by reporting "percentage point increase over the hypothesis-only model”?

Add citations for hypothesis-only models in the context of NLI:
https://aclanthology.org/N18-2017/
https://aclanthology.org/S18-2023/
https://aclanthology.org/L18-1239/

**Reasons To Accept:**

The paper uses LLMs to generate a dataset.

**Reasons To Reject:**

While I am very partial to the idea of analyzing NLI datasets, this paper does not seem relevant to COLM. I fail to see how understanding the NLI task fits under "COLM is an academic venue focused on the study of language modeling, broadly defined, with the goal of creating a community of researchers with expertise in different disciplines, focused on understanding, improving, and critiquing the development of LM technology." I am 100% for "we formulate three
possible readings of the NLI label set and perform a comprehensive analysis of the meta-inferential properties they entail" but this seems to belong at a *ACL or StarSem.

No manual verification/validation is done on the new dataset. While there is justification for the rules used to propagate labels, this should still be verified and reported. What percentage of sampled examples is the propagated label correct/incorrect? How does this breakdown for the different NLI labels?

---

> ### Author Response · Authors · 2025-05-30
>
> Thank you for your review and helpful questions. Please find our responses below.
>
> ## responses to weaknesses
>
> **Relevance to COLM:** We respectfully disagree with the assessment that our paper is not relevant to COLM.  Our core contribution is a methodology to analyze what is learned from an NLI dataset by probing a fine-tuned model for the notion of entailment it encodes. We believe this falls squarely within the scope of the COLM CfP, especially:
>
> > - All about **data**: pre-training data, alignment data, and synthetic data --- via manual or algorithmic analysis, curation, and generation
> > - All about **evaluation**: benchmarks, simulation environments, scalable oversight, evaluation protocols and metrics, human and/or machine evaluation
>
> **Manual verification:** We determined that manual annotation wasn't necessary in the two cases where it could have been performed for the following reasons:
>
> 1. For the _generated items_, BERT trained on the un-augmented SNLI performs very well on the generated test sets (Table 3 row 1). Additionally, models trained on the augmented SNLI perform very well on the original SNLI test set (Table 3 column 1). These two results confirm  that the generated data is non-trivially in-distribution for SNLI with respect to BERT and RoBERTa+SE, which is what we need to assume to perform the subsequent analysis.
>
> 2. For the _propagated items_, we actually don't crate items with ground truth lables, but rather we use the model predictions to draw conclusions about the model's notion of entailment. The same principle could be used to investigate how human speakers approach the NLI task, but that question is out of scope for this paper and may be more suitable for another venue.
>
> ## answers to questions
>
> ### Why is there no RoBERTa baseline trained on just SNLI in [Table 4]?
>
> We chose BERT as a strong baseline because it has been extensively studied in the context of NLI. We chose RoBERTa+SE as it has recently SotA performance on NLI. For a comparison between RoBERTa+SE and RoBERTa we refer the reader to Sut et al. (2020).
>
> ### How exactly does the performance in Table 3 demonstrate that the "generated NLI data adequately replicates the distribution of the human-generated SNLI dataset"?
>
> Row 1 shows that the items are in-distribution for SNLI modulo the BERT encoder since the model trained on SNLI perform well. Column 1 shows that the generated items are not merely trivially within distribution (e.g,. all tautologies or trivial contradictions) since the addition of generated items helps the model to generalize _within_ the original SNLI test set.
>
> ### How many “perfect items” in the SNLI train set are there (line 218)?
>
> There are 36975 re-annotated items in the test set. Of these, 22332 are such that all of the annotators agree on the label. Within these items there are 19 sets of 3 where all of the labels are covered for the same hypothesis (i.e. same `CaptionID`). From these 19 we sampled 10 (30 items). We will clarify this in the text.
>
> ### In 6.1, how can we conclude that BERT "performs about a bit worse on Ll3.2 and DS-R1 and a bit better on Ll3.3 in comparison to the SNLI test set” if the data isnt annotated? Are the labels just propogated based on 234-237? If so, what verification/manual sampling was done to verify these rules for propogating labels.
>
> The analysis in 6.1 does not rely on any propagated labels. This is referring to the first row of Table 3, where BERT is trained on the original SNLI train set and tested on the original SNLI test set (first column) and the LLM-generated test sets (final 3 columns).
>
> ### In [Table 4], in each row, the E, C, N values sum to one. But what are we comparing to? Are about 1/3 of the examples supposed to be E, C, N?
>
> Where the model predictions are _supposed to_ fall depends on the version of entailment that is assumed (columns SC and EI, derived from Table 2). The model's performance _under those different assumptions_ (columns SC$\checkmark$ and EI$\checkmark$) is what we use to make inferences about the version of entailment the model has learned. We will clarify this in the caption to Table 4.
>
> ### What is gained by reporting "percentage point increase over the hypothesis-only model”?
>
> The difference between hypothesis only and full model performance can give an indicator of the severity of the hypothesis-only "informaiton leak" problem. We want to ensure that the LLM-generated hypotheses don't fully give away the entailment relation since that would affect how we can interpret results on the propagated items, where the expected labels can be different from what the hypothesis was originally generated for. As mentioned in §6.1, this is part of the reason we prefer the DS-R1 dataset for the subsequent analysis.
>
> Thank you for bringing the missing references to our attention!

---

> > ### Comment · Reviewer_UYbC · 2025-06-09
> > **On appropriateness for colm**
> >
> > This might be an agree to disagree case that I’m happy to be overruled on

---

> > ### Comment · Reviewer_UYbC · 2025-06-09
> > **Lack of manual verification**
> >
> > I understand the reasons suggested by the authors for not performing manual verification.
> > However, I strongly believe that manual verification is always necessary, even as a simply sanity check. These reasons are not convincing enough to me for not performing even a limited manual verification

---

### Decision · Program_Chairs · 2025-07-08

**Decision:**

Accept

**Comment:**

First, it is my opinion that the paper’s contributions are well within the scope of COLM. LLMs are both fine-tuned and evaluated on NLI-style tasks, so it seems to me that a paper which uses LLMs as a tool to further scientific understanding of NLI should be welcome at this conference. Similarly, the paper draws on formal semantics to organize its analysis and experiments, with the aim of furthering understanding of NLI datasets and model behavior, not as contributions to formal semantics itself. Concerns about the paper’s fit within COLM notwithstanding, overall reviews are positive; the AC recommends the paper for publication at COLM.

The paper offers two primary contributions:

(1) A clarifying discussion of the possible logical interpretations of NLI inference relations, as material implication (MI), strict conditional (SC), and existential import (EI), along with the meta-inferential rules that each interpretation supports.

(2) A thoughtful set of experiments to empirically test which of these interpretations best explains the behavior of models trained on NLI datasets, as well as LLMs prompted in a similar fashion to original NLI instructions.

Suggestions

(1) Reviewers ndSY and UYbC both raise concerns about lack of manual validation/analysis of generated items. Because the data is generated for descriptive analysis purposes rather than prescriptive evaluation purposes, I believe inclusion of a small-scale manual analysis and discussion would be adequate to address this concern.

(2) In the AC’s opinion, the work would benefit from some qualitative analysis or examples. E.g. from Table 4, what is an example of a contradicted pair (a,b) that is neutral when flipped (b,a), i.e. row Cab, column N? Does this example support the hypothesis that the model is taking an EI approach rather than SC? Or could there be some other explanation of the data? (E.g., is it possible that b’s interpretation depends on a in some way such that the meaning of these sentences changes when their order is swapped?)

(3) Lastly, a very relevant recently published paper the authors may wish to include in their discussion: https://aclanthology.org/2025.naacl-long.130/ (This is not a weakness of the current submission since the related work is contemporaneous.)